# Incidence and Impact of Acute Kidney Injury after Liver Transplantation: A Meta-Analysis

**DOI:** 10.3390/jcm8030372

**Published:** 2019-03-17

**Authors:** Charat Thongprayoon, Wisit Kaewput, Natanong Thamcharoen, Tarun Bathini, Kanramon Watthanasuntorn, Ploypin Lertjitbanjong, Konika Sharma, Sohail Abdul Salim, Patompong Ungprasert, Karn Wijarnpreecha, Paul T. Kröner, Narothama Reddy Aeddula, Michael A Mao, Wisit Cheungpasitporn

**Affiliations:** 1Division of Nephrology and Hypertension, Mayo Clinic, Rochester, MN 55905, USA; charat.thongprayoon@gmail.com; 2Department of Military and Community Medicine, Phramongkutklao College of Medicine, Bangkok 10400, Thailand; wisitnephro@gmail.com; 3Division of Nephrology, Beth Israel Deaconess Medical Center, Harvard Medical School, Boston, MA 02215, USA; natthamcharoen@gmail.com; 4Department of Internal Medicine, University of Arizona, Tucson, AZ 85721, USA; tarunjacobb@gmail.com; 5Department of Internal Medicine, Bassett Medical Center, Cooperstown, NY 13326, USA; kanramon@gmail.com (K.W.); ploypinlert@gmail.com (P.L.); drkonika@gmail.com (K.S.); 6Division of Nephrology, Department of Medicine, University of Mississippi Medical Center, MS 39216, USA; sohail3553@gmail.com; 7Clinical Epidemiology Unit, Department of Research and Development, Faculty of Medicine, Siriraj Hospital, Mahidol University, Bangkok 10700, Thailand; p.ungprasert@gmail.com; 8Department of Medicine, Division of Gastroenterology and Hepatology, Mayo Clinic, Jacksonville, FL 32224, USA; karnjuve10@gmail.com (K.W.); thomaskroner@gmail.com (P.T.K.); 9Division of Nephrology, Department of Medicine, Deaconess Health System, Evansville, IN 47747, USA; dr.anreddy@gmail.com; 10Department of Medicine, Division of Nephrology and Hypertension, Mayo Clinic, Jacksonville, FL 32224, USA; mao.michael@mayo.edu

**Keywords:** Acute renal failure, Acute kidney injury, Epidemiology, Incidence, Meta-analysis, Liver Transplantation, Transplantation, Systematic reviews

## Abstract

Background: The study’s aim was to summarize the incidence and impacts of post-liver transplant (LTx) acute kidney injury (AKI) on outcomes after LTx. Methods: A literature search was performed using the MEDLINE, EMBASE and Cochrane Databases from inception until December 2018 to identify studies assessing the incidence of AKI (using a standard AKI definition) in adult patients undergoing LTx. Effect estimates from the individual studies were derived and consolidated utilizing random-effect, the generic inverse variance approach of DerSimonian and Laird. The protocol for this systematic review is registered with PROSPERO (no. CRD42018100664). Results: Thirty-eight cohort studies, with a total of 13,422 LTx patients, were enrolled. Overall, the pooled estimated incidence rates of post-LTx AKI and severe AKI requiring renal replacement therapy (RRT) were 40.7% (95% CI: 35.4%–46.2%) and 7.7% (95% CI: 5.1%–11.4%), respectively. Meta-regression showed that the year of study did not significantly affect the incidence of post-LTx AKI (*p* = 0.81). The pooled estimated in-hospital or 30-day mortality, and 1-year mortality rates of patients with post-LTx AKI were 16.5% (95% CI: 10.8%–24.3%) and 31.1% (95% CI: 22.4%–41.5%), respectively. Post-LTx AKI and severe AKI requiring RRT were associated with significantly higher mortality with pooled ORs of 2.96 (95% CI: 2.32–3.77) and 8.15 (95%CI: 4.52–14.69), respectively. Compared to those without post-LTx AKI, recipients with post-LTx AKI had significantly increased risk of liver graft failure and chronic kidney disease with pooled ORs of 3.76 (95% CI: 1.56–9.03) and 2.35 (95% CI: 1.53–3.61), respectively. Conclusion: The overall estimated incidence rates of post-LTx AKI and severe AKI requiring RRT are 40.8% and 7.0%, respectively. There are significant associations of post-LTx AKI with increased mortality and graft failure after transplantation. Furthermore, the incidence of post-LTx AKI has remained stable over the ten years of the study.

## 1. Introduction

Acute kidney injury (AKI) is associated with high mortality worldwide (1.7 million deaths per year) [1,2,3,4]. Patients who survive AKI are at increased risk for significant morbidities such as hypertension and progressive chronic kidney disease (CKD) [5]. The incidence of AKI has steadily increased in recent years [2]. It has been suggested that AKI’s global burden is 13.3 million cases a year [6]. In the United States, hospitalizations for AKI have been steeply rising, and data from national inpatient sample shows that the number of hospitalizations due to AKI increased from 953,926 in 2000 to 1,823,054 in 2006 and 3,959,560 in 2014, which accounts for one hospitalization associated with AKI every 7.5 minutes [7,8].

AKI is a common and significant complication after liver transplantation (LTx), and is associated with increased mortality, hospital length of stay, utilization of resources, and health care costs [9,10,11,12,13,14,15,16,17,18,19,20,21,22,23,24,25,26,27]. Although the survival of LTx recipients has improved substantially over the past five decades, mortality rates related to post-LTx AKI and subsequent progressive CKD remain high and are of increasing concern [14,15,28,29,30,31]. The underlying mechanisms for post-LTx AKI appear to be complex and differ from other medical or surgery-associated AKI [11,23,24,25,32,33,34,35]. Recent studies have suggested several important factors that influence post-LTx AKI, including hepatic ischemia-reperfusion injury (HIRI) [36,37,38], increased use of high-risk or marginal grafts, and transplantation of liver grafts to sicker patients with higher Model For End-Stage Liver Disease (MELD) score or with more comorbidities [23,39,40,41,42,43,44,45,46,47,48,49,50,51]. In our literature review, the reported incidences are a farrago, having a range between 5% to 94% [10,11,14,15,16,17,18,19,20,21,22,23,24,25,28,29,30,31,32,33,34,35,39,40,41,42,43,44,45,46,47,48,49,52,53,54,55,56,57,58,59,60,61,62,63,64,65,66,67,68,69,70,71,72,73,74,75,76,77,78,79,80]. These wide variabilities are possibly due to non-uniform definitions of AKI [10,11,14,15,16,17,18,19,20,21,22,23,24,25,28,29,30,31,32,33,34,35,39,40,41,42,43,44,45,46,47,48,49,52,53,54,55,56,57,58,59,60,61,62,63,64,65,66,67,68,69,70,71,72,73,74,75,76,77,78,79,80]. In addition, despite progress in transplant medicine, the incidence, risk factors, and mortality associated with AKI in post-LTx patients and their trends remain unclear [10,11,14,15,16,17,18,19,20,21,22,23,24,25,28,29,30,31,32,33,34,35,39,40,41,42,43,44,45,46,47,48,49,52,53,54,55,56,57,58,59,60,61,62,63,64,65,66,67,68,69,70,71,72,73,74,75,76,77,78,79,80,81,82,83].

Thus, we performed a systematic review to summarize the incidence (using standard AKI definitions of Risk, Injury, Failure, Loss of kidney function, and End-stage kidney disease (RIFLE), Acute Kidney Injury Network (AKIN), and Kidney Disease: Improving Global Outcomes (KDIGO) classifications), risk factors, and mortality and their trends for AKI in patients undergoing LTx.

## 2. Methods

### 2.1. Search Strategy and Literature Review

The protocol for this systematic review was registered with PROSPERO (International Prospective Register of Systematic Reviews; no. CRD42018100664). A systematic literature search of MEDLINE (1946 to December 2018), EMBASE (1988 to December 2018) and the Cochrane Database of Systematic Reviews (database inception to December 2018) was performed to evaluate the incidence of AKI in adult patients undergoing LTx. The systematic literature review was conducted independently by two investigators (C.T. and W.C.) using the search strategy that consolidated the terms “acute kidney injury” OR “renal failure” AND “liver transplantation," which is provided in online Appendix A. No language limitation was implemented. A manual search for conceivably related studies using references of the included articles was also performed. This study was conducted by the Preferred Reporting Items for Systematic Reviews and Meta-Analysis (PRISMA) statement [84] and the Strengthening the Reporting of Observational Studies in Epidemiology (STROBE) [85].

### 2.2. Selection Criteria

Eligible studies must be clinical trials or observational studies (cohort, case-control, or cross-sectional studies) that reported the incidence of post-LTx AKI in adult patients (age >/= 18 years old). Included studies must provide data to estimate the incidence of post-LTx AKI with 95% confidence intervals (CI). Retrieved articles were individually reviewed for eligibility by the two investigators (C.T. and W.C.). Discrepancies were addressed and solved by mutual consensus. Inclusion was not limited by the size of study.

### 2.3. Data Abstraction

A structured data collecting form was used to obtain the following information from each study, including title, name of the first author, year of the study, publication year, country where the study was conducted, post-LTx AKI definition, incidence of AKI post-LTx, risk factors for post-LTx AKI, and impact of post-LTx AKI on patient outcomes.

### 2.4. Statistical Analysis

Analyses were performed utilizing the Comprehensive Meta-Analysis 3.3 software (Biostat Inc, Englewood, NJ, USA). Adjusted point estimates from each study were consolidated by the generic inverse variance approach of DerSimonian and Laird, which designated the weight of each study based on its variance [86]. Given the possibility of between-study variance, we used a random-effect model rather than a fixed-effect model. Cochran’s Q test and *I^2^* statistic were applied to determine the between-study heterogeneity. A value of *I^2^* of 0%–25% represents insignificant heterogeneity, 26%–50% low heterogeneity, 51%–75% moderate heterogeneity and 76–100% high heterogeneity [87]. The presence of publication bias was assessed by the Egger test [88].

## 3. Results

A total of 2525 potentially eligible articles were identified using our search strategy. After the exclusion of 1994 articles based on title and abstract for clearly not fulfilling inclusion criteria on the basis of type of article, patient population, study design, or outcome of interest, and 417 due to being duplicates, 114 articles were left for full-length review. Thirty-six of them were excluded from the full-length review as they did not report the outcome of interest, while 17 were excluded because they were not observational studies or clinical trials. Twenty-three studies were subsequently excluded because they did not use a standard AKI definition. Thus, we included 38 cohort studies [14,18,19,21,28,29,30,31,32,39,41,42,43,44,48,49,55,56,57,58,59,60,62,63,64,65,66,69,70,72,73,74,75,76,77,78,79,80] in the meta-analysis of post-LTx AKI incidence with 13,422 patients enrolled. The literature retrieval, review, and selection process are demonstrated in Figure 1. The characteristics of the included studies are presented in Table 1.

### 3.1. Incidence of Post-LTx AKI

Overall, the pooled estimated incidence rates of post-LTx AKI and severe AKI requiring RRT following LTx were 40.7% (95% CI: 35.4%–46.2%, *I*^2^ = 97%, Figure 2) and 7.7% (95% CI: 5.1%–11.4%, *I*^2^ = 95%, Figure 3), respectively.

Meta-regression showed no significant impact of type of donor (deceased vs living donors) (*p* = 0.33) on the incidence of post-LTx AKI. In addition, the year of study (*p* = 0.81) did not significantly affect the incidence of post-LTx AKI (Figure 4).

### 3.2. Risk Factors for Post-LTx AKI

Reported risk factors for post-LTx AKI are demonstrated in Table 2. Higher pretransplant SCr [11,23,24,25,32,33,34,35], high body mass index (BMI) [39,64,66,67], high MELD/MELD-Na score [23,39,40,41,42,43,44,45,46,47,48,49], intraoperative blood loss and perioperative blood transfusion [18,25,39,48,54,65], high APACHE II score [25,43,48,55], hypotension and vasopressor requirement [18,24,48,54], cold and warm ischemia time [14,35,78], graft dysfunction [11,40,53], post-reperfusion syndrome [20,64,66,75,78], infection prior to transplant [25,45,48], and hypoalbuminemia [18,64,66] were consistently identified as important risk factors for Post-LTx AKI.

### 3.3. Impacts of Post-LTx AKI on Patient Outcomes

The impacts of post-LTx AKI on patient outcomes are demonstrated in Table 3. Overall, the pooled estimated in-hospital or 30-day mortality, and 1-year mortality rates of patients with post-LTx AKI were 16.5% (95% CI: 10.8%–24.3%, *I*^2^ = 94%) and 31.1% (95% CI: 22.4%–41.5%, *I*^2^ = 78%), respectively. Post-LTx AKI was associated with significantly higher mortality with a pooled OR of 2.96 (95% CI: 2.32–3.77, *I*^2^ = 59%). In addition, severe post-LTx AKI requiring RRT was associated with significantly higher mortality with a pooled OR of 8.15 (95% CI: 4.52–14.69, *I*^2^ = 90%). Compared to those without post-LTx AKI, recipients with post-LTx AKI had significantly increased risks of liver graft failure and CKD with pooled ORs of 3.76 (95% CI: 1.56–9.03, *I*^2^ = 91%, Figure 5) and 2.35 (95% CI: 1.53–3.61, *I*^2^ = 75%, Figure 6), respectively. AKI was associated with prolonged intensive care (ICU) and hospital stay [17,18,23,24,29,32,35,40,42,44,48,49,53,61,64,75,78] (Table 3).

### 3.4. Evaluation for Publication Bias

Funnel plot (Appendix A) and Egger’s regression asymmetry test were performed to evaluate for publication bias in the analysis evaluating incidence of post-LTx AKI and mortality risk of post-LTx AKI. There was no significant publication bias in meta-analysis assessing the incidence of post-LTx AKI, *p*-value = 0.12.

## 4. Discussion

In this meta-analysis, we found that AKI and severe AKI requiring RRT after LTx are common, with an incidence of 40.8% and 7.0%, respectively. In addition, our findings showed no significant correlation between the incidence of post-LTx AKI and study year for the ten years of the study. Furthermore, compared to patients without post-LTx AKI, those with post-LTx AKI carry a 2.96-fold increased risk of mortality and a 3.76-fold higher risk of liver graft failure.

The development of post-LTx AKI appears to be multifactorial with a number of preoperative, intraoperative and postoperative factors involved [90]. Pre-LTx factors include high MELD/MELD-Na score, high APACHE II score, hypoalbuminemia, and reduced eGFR [11,23,24,25,32,33,34,35]. Preexisting renal impairment is common among patients with end-stage liver disease [91]. Although cirrhotic patients with significant CKD are eligible to receive a combined liver-kidney transplantation [92], a lower baseline GFR among those who received LTx alone remained an important risk factor for post-operative AKI [11,23,24,25,32,33,34,35]. Studies have demonstrated that hepatorenal syndrome before LTx can also lead to renal insufficiency and render LTx recipients more susceptible to post-LTx AKI [22,90,93]. In addition, sepsis, graft dysfunction, thrombotic microangiopathy, and calcineurin inhibitor nephrotoxicity may all contribute to AKI [22,37,94,95,96].

Studies have shown that higher MELD scores were associated with post-LTx AKI [23,39,40,41,42,43,44,45,46,47,48,49]. In patients with high MELD scores >30, the majority required RRT post LTx [44,97]. Although SCr is an important determinant of the MELD score, other components of MELD such as pre-LTx INR has also been demonstrated to be strongly associated with post-LT AKI, suggesting that the severity of the liver disease itself, as reflected by the MELD score, is associated with post-LT AKI [45]. Identified perioperative factors for post-LTx AKI include cardiopulmonary failure, vasopressor requirement, hemodynamic effects of prolonged surgery, and blood loss/RBC transfusion [18,24,25,39,48,54,65]. Moreover, it has been hypothesized that HIRI is an important cause of post-LTx AKI [37,38]. Aspartate aminotransferase (AST), as a surrogate marker for HIRI, has been shown to be correlated with post-LTx AKI. [38,78] HIRI has a close relationship with the systemic inflammatory response, which in turn is related to AKI and multiorgan dysfunction in similar settings such as sepsis [37]. Early hepatic graft dysfunction has also been shown to be associated to post-LTx AKI [98]. In addition, recipients of donation after circulatory death (DCD) grafts are reported to have a higher incidence of post-LTx AKI compared to donation after brain death (DBD grafts). After DCD LTx, peak AST levels were an independent predictor of post-LTx AKI [99]. Other known factors that influence HIRI such as donor age, cold and warm ischemia times and graft steatosis have also been associated with post-LTx AKI [37].

As demonstrated in our study, post-LTx AKI is associated with an increased risk of death and liver graft failure. Several pharmacological and non-pharmacological interventions have been studied, but so far these have failed to demonstrate any significant benefit in the prevention of post-LTx AKI [37,100,101]. To continue efforts to mitigate post-LTx AKI, it is important to identify those who are at high-risk for post-LTx AKI in order to develop earlier protective strategies [37]. There have been many attempts to develop predictive models for post-LTx AKI [37]. Seven published predictive models addressing a diverse range of AKI definitions for post-LT AKI have been developed [19,23,24,33,47,54,55]. However, the numbers of patients in these studies were limited [19,23,24,33,47,54,55], and future prospective external validation, ideally with multi-center studies with large number of patients, is required.

Several limitations in our meta-analysis are worth mentioning. First, there were statistical heterogeneities present in our study. Possible sources for heterogeneities were the differences in the patient characteristics in the individual studies. However, we performed a meta-regression analysis which demonstrated that the type of donor (deceased vs. living donors); the year of study did not significantly affect the incidence of post-LTx AKI. Second, there is a lack of data from included studies on novel AKI biomarkers. Novel biomarkers for AKI are emerging and could be useful for the early identification and characterization of AKI. Thus, future studies evaluating predictive models with novel biomarkers are needed. Lastly, this is a systematic review and meta-analysis of cohort studies. Thus, it can demonstrate associations of post-LTx AKI with increased risk of mortality and liver graft failure, but not a causal relationship.

## 5. Conclusions

In conclusion, there are overall high incidence rates of post-LTx AKI and severe AKI requiring RRT of 40.8% and 7.0%. Post-LTx AKI is significantly associated with increased mortality and liver graft failure. In addition, the incidence of post-LTx AKI has remained stable over time. This study provides an epidemiological perspective to support the need for future large-scale multi-center studies to identify preventive strategies for post-LTx AKI.

## Figures and Tables

**Figure 1 jcm-08-00372-f001:**
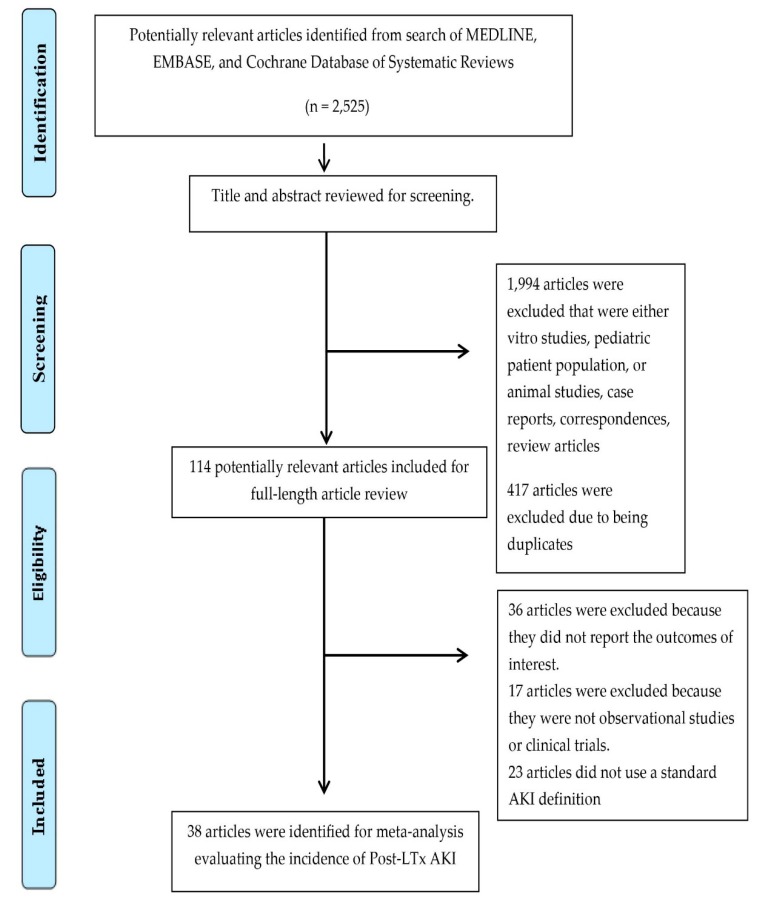
Outline of our search methodology.

**Figure 2 jcm-08-00372-f002:**
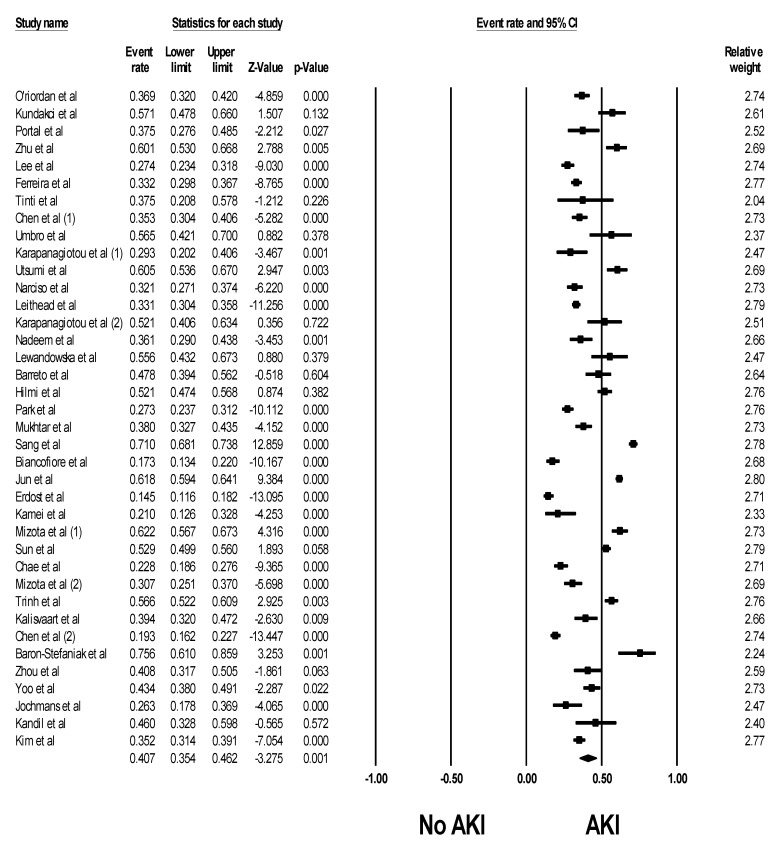
Forest plots of the included studies assessing incidence rates of post-LTx AKI. A diamond data marker represents the overall rate from each included study (square data marker) and 95% confidence interval.

**Figure 3 jcm-08-00372-f003:**
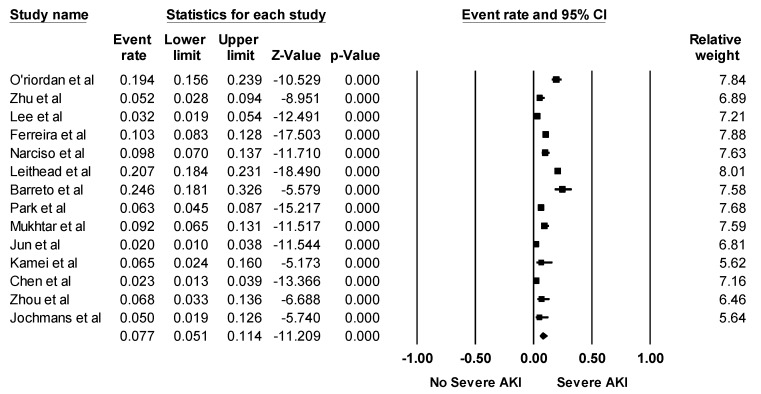
Forest plots of the included studies assessing incidence rates of severe AKI requiring RRT following LTx. A diamond data marker represents the overall rate from each included study (square data marker) and 95% confidence interval.

**Figure 4 jcm-08-00372-f004:**
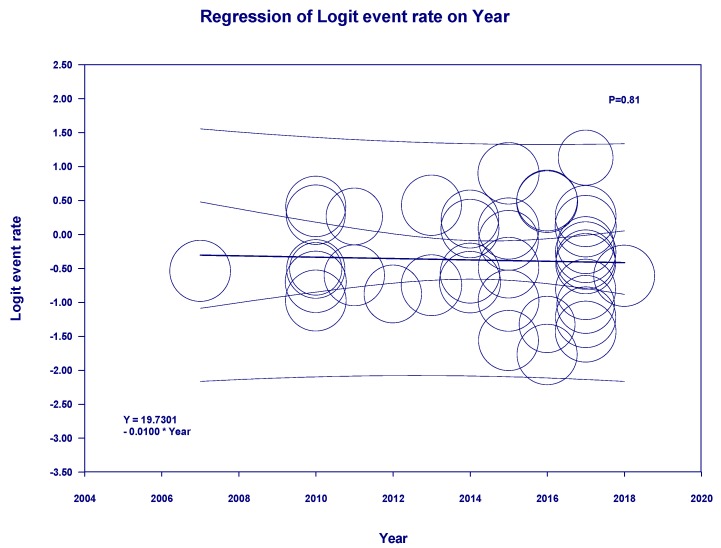
Meta-regression analyses showed no significant impact of year of study on the incidence of post-LTx AKI (*p* = 0.81). The solid black line represents the weighted regression line based on variance-weighted least squares. The inner and outer lines show the 95% confidence interval and prediction interval around the regression line. The circles indicate log event rates in each study.

**Figure 5 jcm-08-00372-f005:**
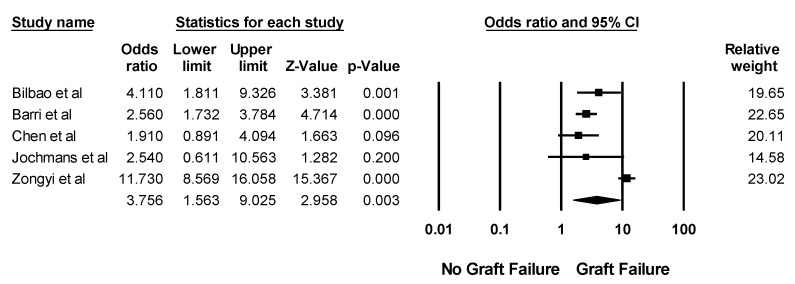
Forest plots of the included studies assessing liver graft failure among patients with post-LTx AKI. A diamond data marker represents the overall rate from each included study (square data marker) and 95% confidence interval.

**Figure 6 jcm-08-00372-f006:**
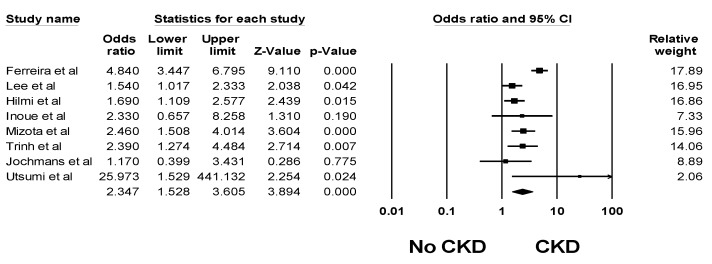
Forest plots of the included studies assessing CKD risk among patients with post-LTx AKI. A diamond data marker represents the overall rate from each included study (square data marker) and 95% confidence interval.

**Table 1 jcm-08-00372-t001:** Main characteristics of studies included in meta-analysis of AKI in patients undergoing LTx [14,18,19,21,28,29,30,31,32,39,41,42,43,44,48,49,55,56,57,58,59,60,62,63,64,65,66,69,70,72,73,74,75,76,77,78,79,80].

Study	Year	Country	Procedure/Patients	Number	Deceased Donor	AKI Definition	Incidence	Mortality in AKI
O’riordan et al. [32]	2007	Ireland	Deceased donor orthotopic liver transplant	350	350 (100%)	ARI/ARF; RIFLE Injury and Failure stage within 2 weeks after transplant	ARI/ARF 129/350 (36.9%) Dialysis 68/350 (19.4%)	1-year mortality 56/129 (43%)
Kundakci et al. [41]	2010	Turkey	Orthotopic liver transplant	112	75 (67%)	AKI; RIFLE criteria	AKI 64/112 (57.1%)	1-year mortality 23/64 (36%)
Portal et al. [55]	2010	UK	Liver transplant	80	N/A	AKI; AKIN criteria within 48 hours after transplants	AKI 30/80 (37.5%)	N/A
Zhu et al. [42]	2010	China	Deceased donor orthotopic liver transplant	193	193 (100%)	AKI; AKIN criteria within 28 days after transplants	AKI 116/193 (60.1%) Dialysis 10/193 (5.2%)	1-year mortality 30/116 (26%)
Lee et al. [56]	2010	Korea	Liver transplant	431	99 (23%)	AKI; RIFLE criteria	AKI 118/431 (27.4%) Dialysis 14/431 (3.2%)	N/A
Ferreira et al. [57]	2010	Portugal	Orthotopic liver transplant	708	N/A	AKI; RIFLE criteria within 21 days after transplant	AKI 235/708 (33.2%) Dialysis 73/708 (10.3%)	Mortality 43/235 (18%)
Tinti et al. [58]	2010	Italy	Deceased donor orthotopic liver transplant	24	24 (100%)	AKI; RIFLE criteria within 15 days after transplant	AKI 9/24 (37.5%)	N/A
Chen et al. (1) [18]	2011	USA	Liver transplant	334	N/A	ARI/ARF; RIFLE Injury and Failure stage within 2 weeks after transplant within 7 days after transplant	ARI/ARF 118/334 (38.3%)	Mortality 13/118 (11%)
Umbro et al. [59]	2011	Italy	Deceased donor liver transplant	46	46 (100%)	AKI; RIFLE criteria within 7 days after transplant	AKI 26/46 (56.5%)	N/A
Karapanagiotou et al. (1) [43]	2012	Greece	Orthotopic liver transplant	75	N/A	AKI; an increase in SCr 1.5 times above baseline or value > 2.0 mg/dL within 7 days after transplant	AKI 22/75 (29.3%) Dialysis 7/75 (9.3%)	1-year mortality 11/22 (50%)
Utsumi et al. [44]	2013	Japan	Living donor liver transplant	200	0 (0%)	AKI; RIFLE criteria within 28 days after transplants	AKI 121/200 (60.5%) ARI/ARF 74/200 (37%)	Hospital mortality AKI 14/121 (12%) ARI/ARF 12/74(16%) 1-year mortality AKI 24/121 (20%) ARI/ARF 22/74 (30%)
Narciso et al. [60]	2013	Brazil	Liver transplant	315	181 (57%)	AKI; AKIN criteria within 48 hours after transplants	AKI 48 hours: 101/315 (32.1%) 1 week: 255/315 (81%) Hospitalization: 293/315 (93%) Dialysis Any: 48/315 (15.2%) 1 week: 31/315 (9.8%)	Dialysis 28/48 (58%)
Leithead et al. [39]	2014	UK	Liver transplant	1152	1152 (100%) DCD 112 (10%)	AKI; KDIGO criteria within 7 days after transplants	AKI 381/1152 (33.1%) Dialysis 238/1152 (20.7%)	AKI 152/381 (40%)
Karapanagiotou et al. (2) [48]	2014	Greece	Liver transplant	71	N/A	AKI; RIFLE within 7 days or AKIN criteria within 48 hours	RIFLE AKI 28/71 (39.4%) AKIN AKI 37/71 (52.1%)	6-month mortality RIFLE AKI 15/28 (54%) AKIN AKI 17/37 (46%)
Nadeem et al. [49]	2014	Saudi Arabia	Liver transplant	158	N/A	AKI; RIFLE criteria within 72 hours after transplants	AKI 57/158 (36.1%)	N/A
Lewandowska et al. [62]	2014	Poland	Orthotopic liver transplant	63	N/A	AKI; RIFLE criteria within 72 hours after transplant	AKI 35/63 (55.6%)	N/A
Barreto et al. [63]	2015	Brazil	Orthotopic liver transplant	134	N/A	AKI; AKIN criteria 2 or 3 within 72 hours after transplants	AKIN stage 2 or 3 64/134 (47.8%) Dialysis 33/134 (24.6%)	N/A
Hilmi et al. [19]	2015	USA	Deceased donor liver transplant	424	424 (100%) EDC 257 (61%)	AKI; KDIGO criteria within 72 hours after transplant	AKI 221/424 (52.1%)	30-day mortality 3/221 (1%)
Park et al. [64]	2015	Korea	Living donor liver transplant	538	0 (0%)	AKI; RIFLE criteria within 30 days after transplant	AKI 147/538 (27.3%) Dialysis 34/538 (6.3%)	Hospital mortality 26/147 (18%) 1-year mortality 29/147 (20%)
Mukhtar et al. [65]	2015	Egypt	Living donor liver transplant	303	0 (0%)	AKI; AKIN criteria within 96 hours after transplant	AKI 115/303 (38%) Dialysis 28/303 (9.2%)	N/A
Sang et al. [66]	2015	Korea	Living donor liver transplant	998	0 (0%)	AKI; RIFLE or AKIN criteria within 7 days after transplant	RIFLE AKI 709/998 (71.0%) AKIN AKI 593/998 (59.4%)	RIFLE AKI 79/709 (11%) AKIN AKI 66/593 (11%)
Biancofiore et al. [69]	2015	Italy	Deceased donor liver transplant	295	295 (100%)	AKI; AKIN criteria within 7 days after transplant	AKIN stage 2 AKI 51/295 (17.3%)	N/A
Jun et al. [70]	2016	Korea	Living donor liver transplant	1617	0 (0%)	AKI; KDIGO criteria within 7 days after transplant	AKI 999/1617 (61.8%) Dialysis 9/448 (2%)	N/A
Erdost et al. [72]	2016	Turkey	Liver transplant	440	194 (44%)	AKI; RIFLE, AKIN, KDIGO criteria within 7 days after transplant	RIFLE AKI 35/440 (8.0%) AKIN AKI 63/440 (14.3%) KDIGO AKI 64/440 (14.5%)	30-day mortality RIFLE AKI 8/35 (23%) AKIN AKI 34/63 (54%) KDIGO AKI 35/64 (55%)
Kamei et al. [73]	2016	Japan	Liver transplant	62	DBD 4 (6%)	AKI; RIFLE injury or failure stage within 4 weeks after transplant	AKI 13/62 (21%) Dialysis 4/62 (6.5%)	N/A
Mizota et al. (1) [74]	2016	Japan	Living donor liver transplant	320	0 (0%)	AKI; KDIGO criteria within 7 days after transplant	AKI 199/320 (62.2%)	Hospital mortality 39/199 (20%)
Sun et al. [21]	2017	USA	Liver transplant	1037	N/A	AKI; AKIN criteria within 48 hours after transplant	AKI 549/1037 (54.9%)	N/A
Chae et al. [75]	2017	Korea	Living donor liver transplant	334	0 (0%)	AKI; AKIN criteria within 48 hours after transplant	AKI 76/334 (22.7%)	Hospital mortality 10/76 (13.2%)
Mizota et al. (2) [76]	2017	Japan	Living donor liver transplant	231	0 (0%)	Severe AKI; KDIGO stage 2 or 3 criteria within 7 days after transplant	Severe AKI 71/231 (30.7%)	Hospital mortality 23/71 (32.4%)
Trinh et al. [77]	2017	Canada	Deceased donor liver transplant	491	491 (100%)	AKI; KDIGO criteria within 7 days after transplant	AKI 278/491 (56.6%)	N/A
Kalisvaart et al. [78]	2017	Netherlands	Donation after brain death liver transplant	155	155 (100%) DBD 155 (100%)	AKI; AKIN criteria within 7 days after transplant	AKI 61/155 (39.4%) Dialysis 5/155 (3.2%)	Hospital mortality 9/61 (15%)
Chen et al. (2) [79]	2017	China	Liver transplant in hepatocellular carcinoma	566	N/A	AKI; AKIN criteria within 48 hours after transplant	AKI 109/566 (19.3%) Dialysis 13/566 (2.3%)	30-day mortality 9/109 (8%)
Baron-Stefaniak et al. [80]	2017	Austria	Orthotopic liver transplant	45	N/A	AKI; KDIGO criteria within 48 hours after transplant	AKI 34/45 (75.6)	N/A
Zhou et al. [30]	2017	China	Donation after circulatory death orthotopic liver transplant	103	103 (100%) DCD 103 (100%)	AKI; KDIGO criteria within 7 days after transplant	AKI 42/103 (40.8%) CRRT 7/103 (6.8%)	N/A
Yoo et al. [31]	2017	Korea	Liver transplant	304	84 (28%)	AKI; RIFLE criteria within 7 days after transplant	AKI 132/304 (43.4%)	N/A
Jochmans [29]	2017	Belgium	Orthotopic liver transplant	80	80 (100%) DCD 13 (16%) DBD 67 (84%)	AKI; RIFLE criteria within 5 days after reperfusion	AKI 21/80 (26.3%) Dialysis 4/80 (5%)	1-year mortality 2/21 (10%)
Kandil et al. [28]	2017	Egypt	Living donor liver transplant	50	0 (0%)	AKI; AKIN criteria within 48 hours	AKI 23/50 (46%)	N/A
Kim et al. [14]	2018	Korea	Living donor liver transplant	583	0 (0%)	AKI; KDIGO criteria within 7 days after transplant	AKI 205/583 (35.2%)	N/A

Abbreviations: AKIN, Acute Kidney Injury Network; DCD, donation after circulatory death; EDC, extended donor criteria liver allografts; KDIGO, Kidney Disease Improving Global Outcomes; RIFLE, Risk, Injury, Failure, Loss of kidney function, and End-stage kidney disease; UK, United Kingdom; USA, United States of America.

**Table 2 jcm-08-00372-t002:** Reported Potential Predictors/Associated-Risk Factors of Post-LTx AKI.

Donor and Graft Factors	Recipient Factors	Surgical and Postoperative Factors
Cold ischemia time [14,35,78], warm ischemic time [35,39,63,64,66] Small-for-size graft/Graft-recipient body weight ratio [40,44,65,66] Deceased donor [20,47] Graft dysfunction [11,53] DCD [39] ABO incompatibility [70] Lower donor BMI [39] Older donor age [39]	Higher MELD score/MELD-Na [23,39,40,41,42,43,44,45,46,47,48,49,64,67,89] APACHE II25 [43,48,55], Preoperative SCr11 [23,24,25,32,33,34,35] Preoperative BUN [23,24] Preoperative renal dysfunction/ARF [40,43,53] Child-Pugh score [19] SOFA [48] Male sex [42], female sex [19,31] Preoperative hepatic encephalopathy [47] Infection [25,48,71] Hypoalbuminemia [18,53,64,66] Preoperative low hemoglobin [14,72] High body weight, BMI [14,19,39,44,64,66,67,75] Pretransplant hypertension [32,54] Preoperative DM [19,44] Alcoholic liver disease [32] Pretransplant hepatitis B and/or C [54,63] Tumor as indication for transplant [47] Elevated lactate [54,63] Elevated plasma NGAL [55] Hyponatremia [39] Pulmonary hypertension [31]	Intra-operative hypotension, low MAP [24,33,34,54,66,79] Inotrope/vasopressor requirement [18,30,32,48,65], dopamine [35], intra-operative need of noradrenaline [33,67] Duration of treatment with dopamine [53] Blood loss [35,44,47,64,70,71], RBC transfusion [14,18,25,33,39,48,54,65,66,72,89] Need of cryoprecipitate [64] Anesthetic/Operation time [30,64,66,70] Post-reperfusion syndrome [20,64,66,78] SvO2 reduction with oliguria [14], Oxygen content 5 min after graft reperfusion [75] Terlipressin (protective) [65] Venovenous bypass (protective) [21] Postoperative ICU days [23,48] Duration of ventilator support [48] Aminoglycoside use [32] Duration of anhepatic phase [41,79] Intra-operative acidosis [41] Intra-operative urine output [14,24,30,33] Overexposure to calcineurin inhibitor [35,44,64] Need of diuretics [46,75] Chloride-liberal fluid received within the 24 h posttransplant [49] Crystalloid administration [14] Use of 6% HES [89] Mean blood glucose during the day of surgery [64], glucose variability [31] Peak AST occurring at 6 h [29]

Abbreviations:: ABO incompatibility, incompatibility of the ABO blood group; AKI, acute kidney injury; AKIN, Acute Kidney Injury Network; ALP, alkaline phosphatase; APACHE, Acute Physiology and Chronic Health Evaluation; ARI, acute renal injury; ARF, acute renal failure; AST, aspartate aminotransferase; ATG, Anti-thymocyte globulin; BMI, body mass index; BUN, blood urea nitrogen; CMV, cytomegalovirus; DBD, graft donated after brain death; DCD, donation after circulatory death; DM, diabetes mellitus; eGFR, estimated glomerular filtration rate; FFP, fresh frozen plasma; HCV, hepatitis C virus; HES, hydroxyethyl starch; ICU, intensive care unit; KDIGO, Kidney Disease Improving Global Outcomes; SCr, serum creatinine; MAP, mean arterial pressure; MELD, Model For End-Stage Liver Disease; MMF, mycophenolate mofetil; N/A, not available; NGAL, neutrophil gelatinase-associated lipocalin; PBC, primary biliary cirrhosis; RBC, red blood cell; RRT, renal replacement therapy; RIFLE, Risk, Injury, Failure, Loss of kidney function, and End-stage kidney disease; SOFA, Sequential Organ Failure Assessment; SvO2, mixed venous oxygen saturation.

**Table 3 jcm-08-00372-t003:** Reported Outcomes of Post-LTx AKI.

Study	Outcomes	Confounder Adjustment
Bilbao et al. [11]	Mortality Dialysis: 6.47 (2.73–15.35) Graft failure Dialysis: 4.11 (1.81–9.32)	None
Contreras et al. [24]	Hospital mortality Dialysis: 9.91 (3.45–28.51) ICU LOS Dialysis: 15 ± 13 vs. 7 ± 11 days Hospital LOS Dialysis: 34 ± 27 vs. 19 ± 20 days	None
Lebrón Gallardo et al. [25]	Mortality Early renal dysfunction: 2.47 (1.29–4.72) Dialysis: 8.80 (3.65–21.23)	None
Sanchez et al. [23]	1-year mortality Dialysis: 9.07 (5.49–14.97) ICU LOS 2.1 ± 3.0 in no dialysis vs. 8.6 ± 11.6 in hemodialysis vs. 10.5 ± 12.8 days in CRRT	None
Wyatt et al. [22]	Mortality ARF without RRT: 8.69 (3.25–23.19) ARF with RRT: 12.07 (3.90–37.32)	Age, sex, race, DM, transplant centers
Cabezuelo et al. [53]	ICU LOS ARF: 12.9 ± 7.4 vs. 7.2 ± 4.0 days	N/A
O’Riordan et al. [32]	1-year mortality ARF: 2.6 (1.5–4.5) Hospital LOS 39.3 ± 79.5 in no ARI/ARF vs. 53.3 ± 72.8 in ARI vs. 73.0 ± 129.8 days in ARF	DM, pretransplant, SCr, PBC, inotrope use, CMV infection/disease, rejection
Lee et al. [40]	Hospital LOS Renal dysfunction: 75 ± 144 vs. 45.2 ± 34.5 days	N/A
Rueggeberg et al. [54]	1-year mortality AKI: 10.93 (3.64–32.83)	None
Barri et al. [17]	2-year mortality AKI: 2.33 (1.53–3.53) 2-year graft failure AKI: 2.56 (1.73–3.78) ICU LOS AKI: 8 ± 19 vs. 3 ± 5 days Hospital LOS AKI: 20 ± 24 vs. 11 ± 10 days	None
Kundakci et al. [41]	1-year mortality AKI: 6.73 (2.15–21.06)	None
Zhu et al. [42]	1-year mortality AKI: 12.1 (1.57–93.54) ICU LOS AKI: 6 (4–9) vs. 4 (3–5) days Hospital LOS AKI: 29 (16–47) vs. 29 (20–48) days	Hypertension, infection and APACHE II
Ferreira et al. [57]	Mortality AKI: 0.73 (0.59–1.08) CKD AKI: 4.84 (3.45–6.80)	None
Lee et al. [56]	CKD AKI: 1.54 (1.02–2.34)	Age, sex, period of transplant, BMI, pretransplant DM, pretransplant hypertension, history of cardiovascular disease, donor type, underlying liver disease, HBV-related liver disease, hepatocellular carcinoma, use of adefovir, calcineurin inhibitors, purine metabolism inhibitors, acute rejection, pretransplant hemoglobin, pretransplant GFR, pretransplant proteinuria, hepatorenal syndrome, Child-Pugh score, MELD score
Chen et al. [18]	1-year mortality ARI/ARF: 2.79 (0.96–8.12) 1-year graft failure ARI/ARF: 1.91 (0.89–4.09) Hospital LOS 21.8 ± 22.1 in no ARI/ARF vs. 24 ± 25 in ARI and 37 ± 49 days in ARF	None
Karapanagiotou et al. [43]	1-year mortality 9.61 (1.48–62.55)	Infection, hemorrhage, MELD, APACHE score
Utsumi et al. [44]	Hospital mortality AKI: 5.04 (1.11–22.81) ARI/ARF: 5.90 (1.83–19.06) 1-year mortality AKI: 9.53 (2.18–41.56) ARI/ARF: 12.90 (4.24–39.30) CKD AKI: 15/107 (14%) vs. 0/77 (0%) ARI/ARF: 35.29 (4.51–275.82) Hospital LOS ARI/ARF: 101.5 ± 68.8 vs. 69.7 ± 48.5 days	None
Narciso et al. [60]	Mortality Dialysis: 6.7 (3.49–12.96)	None
Romano et al. [45]	Hospital mortality AKI: 1.88 (0.76–4.65)	None
Leithead et al. [39]	Mortality 1.71 (1.35–2.17)	Age, sex, MELD score, eGFR, DM
Klaus et al. [46]	Mortality AKI: 5.11 (1.39–18.71) Dialysis:14.4 (4.60–45.09)	None
Kim et al. [47]	1-year mortality Dialysis: 56.5 (12.32–259.20)	None
Karapanagiotou et al. [48]	6-month mortality RIFLE: 3.08 (1.09–1.95) AKIN: 9.34 (1.20–15.69) ICU LOS RIFLE: 15.44 ± 15.41 vs. 8.65 ± 12.59 days AKIN: 13.75 ± 14.53 vs. 9.1 ± 13.08 days	Vasopressor use, RBC transfusion
Nadeem et al. [49]	ICU LOS AKI: 13.4 ± 19 vs. 5.5 ± 4.7 days	N/A
Kirnap et al. [61]	Mortality AKI: 1.85 (0.65–5.23) ICU LOS AKI: 10 ± 8 vs. 3 ± 2 days Hospital LOS AKI: 26 ± 70 vs. 16 ± 7 days	None
Barreto et al. [63]	Hospital mortality AKIN stage 2 or 3: 4.3 (1.3–14.6)	None
Hilmi et al. [19]	30-day mortality AKI: 3/221(1.4%) vs. 0/203 (0%) CKD AKI: 1.69 (1.11–2.58)	None
Park et al. [64]	Hospital mortality 3.44 (1.89–6.25) 1-year mortality AKI: 1.57 (0.95–2.58) ICU LOS 6 (6–7) in no AKI vs. 6 (6–9) in Risk vs. 7 (6–18) in Injury vs. 11 (10–85) in Failure group Hospital LOS 29 (23–42) in no AKI vs. 31 (21–43) in Risk vs. 33 (26–47) in Injury vs. 46 (16–108) in Failure group	None
Mukhtar et al. [65]	Mortality AKI: 2.1 (1.18–4.0)	Graft weight to recipient body weight ratio, baseline creatinine, MELD score, DM, Terlipressin use, massive transfusion, vasopressor use
Sang et al. [66]	Mortality RIFLE AKI: 2.29 (1.29–4.05) AKIN AKI: 1.69 (1.06–2.67)	None
Wyssusek et al. [67]	Mortality AKI: 3.23 (0.43–24.27)	None
Jun et al. [70]	Mortality AKI: 0.36 (0.09–1.43)	ABO incompatibility, MELD score, hypertension, coronary artery disease, age, post-reperfusion syndrome, vasopressor, crystalloid, RBC transfusion, FFP transfusion, operation time, cold ischemic time
Inoue et al. [71]	1-year mortality AKI: 4.54 (1.27–16.32) CKD AKI: 2.33 (0.66–8.29)	None
Mizota et al. [74]	Hospital mortality AKI: 2.53 (1.23–5.22) CKD AKI: 2.46 (1.51–4.02)	Age, MELD score, blood type incompatibility, re-transplantation
Erdost et al. [72]	30-day mortality RIFLE AKI: 4.15 (1.72–10.00) AKIN AKI: 440.83 (58.24–3336.87) KDIGO AKI: 35/64 (55%) vs. 0/376	None
Chae et al. [75]	Hospital mortality AKI: 1.63 (0.73–3.60) ICU LOS AKI: 7 (6–8) vs. 7 (5–7) days Hospital LOS AKI: 28 (22–39) vs. 23 (21–31) days	None
Mizota et al. [76]	Hospital mortality Severe AKI: 3.56 (1.78–7.09)	None
Trinh et al. [77]	Mortality AKI: 1.41 (1.03–1.92) CKD stage 4–5 AKI: 2.39 (1.27–4.47)	Age, sex, MELD score, baseline eGFR, ATG induction, pretransplant hypertension and DM
Kalisvaart et al. [78]	Hospital mortality AKI: 7.96 (1.66–38.25) ICU LOS AKI: 3 (2–5) vs. 2 (2–3) days Hospital LOS AKI: 24 (19–35) vs. 17 (14–27) days	None
Nadkarni et al. [16]	Hospital mortality Dialysis: 2.00 (1.55–2.59)	Not specified
Chen et al. [79]	30-day mortality AKI: 4.05 (1.02–16.18)	ALP, MELD score, operation time, blood transfusion
Zongyi et al. [35]	1-year mortality RIFLE failure stage AKI: 12.25 (8.99–16.70) 1-year graft failure RIFLE failure stage AKI: 11.73 (8.57–16.06) Hospital LOS RIFLE failure stage AKI: 16 (6–34.5) vs. 25 (18–35) days	None
Zhou et al. [30]	14-day mortality AKI: 3.35 (0.94–11.98) Hospital LOS AKI: 28.13 ± 20.04 vs. 26.16 ± 11.91 days	None
Jochmans et al. [29]	1-year mortality AKI: 6.11 (0.52–71.16) 1-year graft failure AKI: 2.54 (0.61–10.55) CKD AKI:1.17 (0.40–3.44) ICU LOS AKI: 4 (3–9) vs. 2 (2–4) Hospital LOS AKI: 23 (17–46) vs. 16 (13–26)	None

Abbreviations:: ABO incompatibility, incompatibility of the ABO blood group; AKI, acute kidney injury; AKIN, Acute Kidney Injury Network; ALP, alkaline phosphatase; APACHE, Acute Physiology and Chronic Health Evaluation; ARI, acute renal injury; ARF, acute renal failure; AST, aspartate aminotransferase; ATG, Anti-thymocyte globulin; BMI, body mass index; BUN, blood urea nitrogen; CMV, cytomegalovirus; DCD, donation after circulatory death; DM, diabetes mellitus; eGFR, estimated glomerular filtration rate; FFP, fresh frozen plasma; HCV, hepatitis C virus; HES, hydroxyethyl starch; ICU, intensive care unit; KDIGO, Kidney Disease Improving Global Outcomes; SCr, serum creatinine; MAP, mean arterial pressure; MELD, Model For End-Stage Liver Disease; MMF, mycophenolate mofetil; N/A, not available; NGAL, neutrophil gelatinase-associated lipocalin; PBC, primary biliary cirrhosis; RBC, red blood cell; RRT, renal replacement therapy; RIFLE, Risk, Injury, Failure, Loss of kidney function, and End-stage kidney disease; SOFA, Sequential Organ Failure Assessment; SvO2, mixed venous oxygen saturation.

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
