# Peer review of "Incidence and Impact of Acute Kidney Injury after Liver Transplantation: A Meta-Analysis"

_jcm, 2019, doi:10.3390/jcm8030372_

Reviewer 1 Report

Thongprayoon and colleagues have performed a systematic review and meta-analysis of the incidence of AKI post liver transplant and its association with outcome. The incidence of AKI is varied across the literature and therefore this pooling of AKI incidence is of value. Its association with increased mortality is similarly of value. The review is well designed and the analysis is sound.

It is surprising that the use of a DCD graft is not associated with and increased incidence of AKI as several studies have shown a strong correlation between the use of a DCD graft and the incidence of AKI. Could the authors comment on how many studies included DCD and DBD grafts and the numbers involved?

Did the authors collect any data regarding AKI and the length of hospital/ITU stay?

There are some minor grammar corrections scattered throughout the article that will need addressed.

Author Response

Reviewer #1
"Thongprayoon and colleagues have performed a systematic review and meta-analysis of the incidence of AKI post liver transplant and its association with outcome. The incidence of AKI is varied across the literature and therefore this pooling of AKI incidence is of value. Its association with increased mortality is similarly of value. The review is well designed and the analysis is sound.

Response: We thank you for reviewing our manuscript and for your critical evaluation. We really appreciated your input and found your suggestions very helpful.

Comment#1  It is surprising that the use of a DCD graft is not associated with and increased incidence of AKI as several studies have shown a strong correlation between the use of a DCD graft and the incidence of AKI. Could the authors comment on how many studies included DCD and DBD grafts and the numbers involved?

Response: We appreciated the reviewer’s important input and observation. The reviewer raised very important point. We apologize for being unclear on analysis of type of donor. Our meta-analysis showed no significant impact of type of donor (deceased vs living donors) (p=0.33) on the incidence of post-LTx AKI. However, the data on DCD graft were limited to perform the meta-analysis. As the reviewer’s suggestion, we have now clarified this point in our results and discussion section. Also, we respected the reviewer’s input and have added data on (%)deceased donors and (%)DCD (when data available) from all included studies in Table 1. In addition, we emphasized that DCD graft is an important associated factor for LT-AKI in Table 2. In addition, we also emphasized this important point in the discussion of the manuscript:

 “Furthermore, recipients of donation after circulatory death (DCD) grafts are reported to have a higher incidence of post-LTx AKI, compared to recipients with graft donated after brain death (DBD grafts). After DCD LTx, peak AST levels were an independent predictor of post-LTx AKI”
Comment#2  Did the authors collect any data regarding AKI and the length of hospital/ICU stay?

Response: We agree with the reviewer. As the reviewer’ suggestion, we have reviewed all included studies have added the data regarding AKI and the length of hospital/ICU stay. “AKI was associated with prolonged intensive care (ICU) and hospital stay10,21,22,27,29,32,35,40,42,44,48,49,53,61,64,75,78 as shown in Table 3.

In addition, we also have added the data on post-LTx AKI and CKD.

Comment#3  There are some minor grammar corrections scattered throughout the article that will need addressed

Response: We appreciated the reviewer’s input. We apologize for minor errors. We have reviewed the manuscript throughout the manuscript for grammar corrections with the coauthors who are native English users.  

All authors thank the Editors and reviewers for their valuable suggestions. The manuscript has been improved considerably by the suggested revisions! 

Reviewer 2 Report

The authors have done a thorough review and meta-analysis of acute kidney injury (AKI) after liver transplantation. Multiple studies (mainly retrospective cohort studies) have been conducted and a comprehensive analysis is more than welcome. The review contains a large population of more than 13.000 patients and AKI was present in 41%. AKI was both related to increased mortality and liver graft failure after liver transplantation.

I have several comments:

·        The introduction is fairly long, please leave out the general data about the incidence of AKI in general and liver diseases treated with transplantation and how many liver transplants are performed. I would rather see a synopsis of the specific etiology of AKI after liver transplantation, which is completely different from AKI in general or AKI after other major surgeries. The results show that the incidence of AKI did not decreased over time, despite medical developments in immunosuppression and surgery. From recent studies and our own experience, this might be due to the increased use of (1) high-risk or marginal grafts, such as grafts from older donors, steatotic grafts and DCD grafts, and (2) liver grafts given to sicker patients (high MELD) or with more comorbidities (older patients, NASH). I would rather have these trends pointed out in the introduction and discussed in the discussion.

·        The authors investigated the ‘impact’ of AKI on mortality and graft failure. I wonder if AKI itself has an impact on these long-term outcomes, or is AKI just a symptom of an early postoperative course seen often with high-risk grafts or high-risk patients. It does not mean that AKI after liver transplantation is not important, it could potentially be used as a predictor for these long-term outcomes and chronic kidney disease. Can the authors please discuss this

·        I suggest the authors ask for the input of a transplant surgeon or hepatologist regarding the clinical course and implications of acute kidney injury after liver transplantation. I have seen that nor a surgeon or hepatologist is an author on this manuscript.

·        The results show type of donor did not impact on the development of AKI. What type of donor do they mean? Living, deceased, DCD, DBD? Please describe.

·        Could the authors please include into table 1 the time period that studies defined in which AKI could develop? I suspect there would be a large variation in this.

·        Could the authors summarize the risk factors in table 2 in a more comprehensive way? i.e. donor, graft, patient, surgical and postoperative factors?

·        An additional analysis (meta-analysis or systematic review) of the relation between AKI and chronic kidney disease would be very interesting. I understand not all studies will have investigated this, but the relation between early and late renal problems after liver transplants would be of great additional value to the paper.

Author Response

Reviewer #2 The authors have done a thorough review and meta-analysis of acute kidney injury (AKI) after liver transplantation. Multiple studies (mainly retrospective cohort studies) have been conducted and a comprehensive analysis is more than welcome. The review contains a large population of more than 13.000 patients and AKI was present in 41%. AKI was both related to increased mortality and liver graft failure after liver transplantation. I have several comments.

Response: We thank you for reviewing our manuscript and for your critical evaluation. We really appreciated your input and found your suggestions very helpful.

Comment#1  The introduction is fairly long, please leave out the general data about the incidence of AKI in general and liver diseases treated with transplantation and how many liver transplants are performed. I would rather see a synopsis of the specific etiology of AKI after liver transplantation, which is completely different from AKI in general or AKI after other major surgeries. The results show that the incidence of AKI did not decreased over time, despite medical developments in immunosuppression and surgery. From recent studies and our own experience, this might be due to the increased use of (1) high-risk or marginal grafts, such as grafts from older donors, steatotic grafts and DCD grafts, and (2) liver grafts given to sicker patients (high MELD) or with more comorbidities (older patients, NASH). I would rather have these trends pointed out in the introduction and discussed in the discussion.

Response: The reviewer is very thorough and has made very good point. We appreciated the reviewer input. We agree with the reviewer and thus we removed the paragraphs on general data about the incidence of AKI in general and liver diseases treated with transplantation. We respected the reviewer and have revised the introduction as the reviewer’s suggestion.

Comment#2 The authors investigated the ‘impact’ of AKI on mortality and graft failure. I wonder if AKI itself has an impact on these long-term outcomes, or is AKI just a symptom of an early postoperative course seen often with high-risk grafts or high-risk patients. It does not mean that AKI after liver transplantation is not important, it could potentially be used as a predictor for these long-term outcomes and chronic kidney disease. Can the authors please discuss this

Response: We appreciate the reviewer’s input. We agree with the review and we have reviewed all included studies have added the data regarding AKI after liver transplantation and CKD risk. We included the data on CKD risk on Table 3 and have performed additional analysis on CKD risk (Figure 6). The following texts have been added in the manuscript

Compared to those without post-LTx AKI, recipients with post-LTx AKI had significant risks of liver graft failure and CKD with pooled ORs of 3.76 (95%CI: 1.56-9.03, I2 = 91%, Figure 5) and 2.35 (95%CI: 1.53-3.61, I2 = 75%, Figure 6), respectively.”

Comment#3  I suggest the authors ask for the input of a transplant surgeon or hepatologist regarding the clinical course and implications of acute kidney injury after liver transplantation. I have seen that nor a surgeon or hepatologist is an author on this manuscript.

Response: We appreciate the reviewer’s input. We apologize for incomplete affiliations of authors. As the reviewer’s suggestion, we have included input from co-authors from Division of Gastroenterology and Hepatology, Mayo Clinic, Jacksonville.

Comment#4 The results show type of donor did not impact on the development of AKI. What type of donor do they mean? Living, deceased, DCD, DBD? Please describe.

Response: We appreciated the reviewer’s important input and observation. The reviewer raised very important point. We apologize for being unclear on analysis of type of donor. Our meta-analysis showed no significant impact of type of donor (deceased vs living donors) (p=0.33) on the incidence of post-LTx AKI. However, the data on DCD graft were limited to perform the meta-analysis. As the reviewer’s suggestion, we have now clarified this point in our results and discussion section. Also, we respected the reviewer’s input and have added data on (%)deceased donors and (%)DCD (when data available) from all included studies in Table 1. In addition, we emphasized that DCD graft is an important associated factor for LT-AKI in Table 2. In addition, we also emphasized this important point in the discussion of the manuscript:

 “Furthermore, recipients of donation after circulatory death (DCD) grafts are reported to have a higher incidence of post-LTx AKI, compared to recipients with graft donated after brain death (DBD grafts). After DCD LTx, peak AST levels were an independent predictor of post-LTx AKI”

Comment#5  Could the authors please include into table 1 the time period that studies defined in which AKI could develop? I suspect there would be a large variation in this.

Response: We appreciate the reviewer’s input. We agree with the reviewer’s important point. We reviewed all included studies have added the data on time period that studies defined in which AKI could develop in revised Table 1.

Comment#6  Could the authors summarize the risk factors in table 2 in a more comprehensive way? i.e. donor, graft, patient, surgical and postoperative factors?

Response: We agree with the reviewer. Thus, we created the new revised Table 2 to summarize reported risk factors as the reviewer’s suggestion.

Comment#7  An additional analysis (meta-analysis or systematic review) of the relation between AKI and chronic kidney disease would be very interesting. I understand not all studies will have investigated this, but the relation between early and late renal problems after liver transplants would be of great additional value to the paper.

Response: We agree with the reviewer. Thus, we reviewed all included studies have added the data regarding AKI after liver transplantation and CKD risk. We included the data on CKD risk on Table 3 and have performed additional analysis on CKD risk (Figure 6).

All authors thank the Editors and reviewers for their valuable suggestions. The manuscript has been improved considerably by the suggested revisions! 

Reviewer 3 Report

Overall, I found the Meta-analysis helpful for all physicians caring for liver transplant patients as it includes a large number of patients from transplants centers throughout the world and different patient populations, surgical approaches, and donor organs. Thus the conclusions that there is a significant incidence of kidney disease and associated mortality after liver transplantation and that there is no one proven modality or treatment protocol that prevents particularly sobering and strong.

I feel that the paper should be published but I would like to suggest certain editorial changes:

Line 46: I suggest rewriting the sentence that begins with Furthermore to be: Furthermore, the incidence of post-LTx AKI has remained stable over the ten years of the study.

Lines 51-61: I suggest eliminating the first paragraph, which I did not feel added anything to the paper nor was that information used later on.

Line 65: Please change "etiopathogenesis" to "etiopathogeneses" and "diseases" to "disease".

Line 71: Please remove "the" prior to LTx and add "is" prior to associated.

Line 72: Please eliminate "and high" prior to utilization, place a "," after resources, and make "cost" plural "costs".

Line 73: I suggest rewriting the start of the second sentence to highlight the survival rates by saying, "The survival of LTx recipients has improved substantially over the past five decades,".

Line 74: Change "is" to "are".

Line 75:  I did not know what "farrago" means and had to look it up; I learned it means hodge-podge or mish-mash. The importance of the sentence I feel would be better expressed by rewriting it as follows, "On literature review, the reported incidences are a farrago having a wide range between 5% to 94%".

Line 77: Please eliminate "used in these studies".

Line 77: The sentence that begins with, "In addition" and ends with "remain unclear", is confusing. There is the nouns "incidence" and "incidence trends" which confuse me. Would the sentence be better written as, "In addition, despite progress in transplant medicine, the incidence, the risk factors, and the mortality associated with AKI in post-LTx patients and their trends remain unclear.".

Line 211: Consider eliminating the phrase starting with "representing" and have the sentence read, "In addition, our findings showed no significant correlation between the incidence of post-LTx AKI and the study year for the ten years of the study."

Line 219: When I read the cited article, the UNOS regulations are that for patients being listed for a SLK, they need to have had evidence of CKD or a GFR less than or equal to 60 for 90 days and a GFR less than or equal to 30 or being on dialysis at the time of listing. So, the criteria are stricter than indicated in the article. 

Author Response

Reviewer #3
Overall, I found the Meta-analysis helpful for all physicians caring for liver transplant patients as it includes a large number of patients from transplants centers throughout the world and different patient populations, surgical approaches, and donor organs. Thus the conclusions that there is a significant incidence of kidney disease and associated mortality after liver transplantation and that there is no one proven modality or treatment protocol that prevents particularly sobering and strong.

I feel that the paper should be published but I would like to suggest certain editorial changes:

Response: We thank you for reviewing our manuscript and for your critical evaluation. We really appreciated your input and found your suggestions very helpful.

Comment#1

Line 46: I suggest rewriting the sentence that begins with Furthermore to be: Furthermore, the incidence of post-LTx AKI has remained stable over the ten years of the study.

Response: We agree with the reviewer. As the reviewer’s suggestion, we have re-written the sentence that begins with Furthermore to be: Furthermore, the incidence of post-LTx AKI has remained stable over the ten years of the study.

Comment#2

Lines 51-61: I suggest eliminating the first paragraph, which I did not feel added anything to the paper nor was that information used later on.

Response: We agree with the reviewer. As the reviewer’s suggestion, we have deleted the first paragraph of the introduction.

Comment#3

Line 65: Please change "etiopathogenesis" to "etiopathogeneses" and "diseases" to "disease".

Line 71: Please remove "the" prior to LTx and add "is" prior to associated.

Response: We agree with the reviewer. As the reviewer’s suggestion, we have made changes accordingly.

Comment#4

Line 72: Please eliminate "and high" prior to utilization, place a "," after resources, and make "cost" plural "costs".

Response: We agree with the reviewer. As the reviewer’s suggestion, we have made changes accordingly.

Comment#5

Line 73: I suggest rewriting the start of the second sentence to highlight the survival rates by saying, "The survival of LTx recipients has improved substantially over the past five decades,".

Response: We agree with the reviewer. We appreciated the reviewer’s input. As the reviewer’s suggestion, we have made changes accordingly.

Comment#6

Line 74: Change "is" to "are".

Response: We agree with the reviewer. As the reviewer’s suggestion, we have made changes accordingly.

Comment#7

Line 75:  I did not know what "farrago" means and had to look it up; I learned it means hodge-podge or mish-mash. The importance of the sentence I feel would be better expressed by rewriting it as follows, "On literature review, the reported incidences are a farrago having a wide range between 5% to 94%".

Response: We agree with the reviewer. As the reviewer’s suggestion, we have made changes accordingly.

Comment#8

Line 77: Please eliminate "used in these studies".

Response: We agree with the reviewer. As the reviewer’s suggestion, we removed "used in these studies".

Comment#9

Line 77: The sentence that begins with, "In addition" and ends with "remain unclear", is confusing. There is the nouns "incidence" and "incidence trends" which confuse me. Would the sentence be better written as, "In addition, despite progress in transplant medicine, the incidence, the risk factors, and the mortality associated with AKI in post-LTx patients and their trends remain unclear.".

Response: We agree with the reviewer. As the reviewer’s suggestion, we have made changes accordingly.

Comment#10

Line 211: Consider eliminating the phrase starting with "representing" and have the sentence read, "In addition, our findings showed no significant correlation between the incidence of post-LTx AKI and the study year for the ten years of the study."

Response: We agree with the reviewer. As the reviewer’s suggestion, we have made changes accordingly.

Comment#11

Line 219: When I read the cited article, the UNOS regulations are that for patients being listed for a SLK, they need to have had evidence of CKD or a GFR less than or equal to 60 for 90 days and a GFR less than or equal to 30 or being on dialysis at the time of listing. So, the criteria are stricter than indicated in the article.

Response: We agree with the reviewer. The reviewer raised very important point. We apologize and agree with the reviewer that the UNOS regulations’ criteria are stricter (https://optn.transplant.hrsa.gov/media/1192/0815-12_SLK_Allocation.pdf Page 5).

Since the regulations/Policy may change or be updated overtime, we have removed the sentence “(e.g., GFR 60 mL/min for longer than 90 consecutive days in the United States) from our manuscript.

All authors thank the Editors and reviewers for their valuable suggestions. The manuscript has been improved considerably by the suggested revisions! 

Round  2

Reviewer 2 Report

I thank the Authors for their revision and accept the manuscript.